# Influential Factors on Postgraduates’ Satisfaction with the Quality of Pharmacy Education: Evidence from a University in Vietnam

**DOI:** 10.3390/pharmacy13030062

**Published:** 2025-04-29

**Authors:** Do Xuan Thang, Nguyen Viet Hung, Vu Tran Anh, Vu Thi Quynh Mai, Le Thu Thuy, Cuc Thi Thu Nguyen, Trung Nguyen Duc, Dang Viet Hung

**Affiliations:** 1Faculty of Pharmaceutical Management and Economics, Hanoi University of Pharmacy, Hanoi 100000, Vietnam; mai.vtq@st.buv.edu.vn (V.T.Q.M.); lethuy@hup.edu.vn (L.T.T.); cucnguyen.pharm@gmail.com (C.T.T.N.); 2Office of Academic Affairs, Hanoi University of Pharmacy, Hanoi 100000, Vietnam; hungnv@hup.edu.vn; 3Department of Quality Assurance and Assessment, Hanoi University of Pharmacy, Hanoi 100000, Vietnam; anhvt@hup.edu.vn; 4Faculty of Pharmacy, The 108 Military Central Hospital, Hanoi 100000, Vietnam; trungnd@benhvien108.vn; 5University Council, Hanoi University of Pharmacy, Hanoi 100000, Vietnam; hungdv@hup.edu.vn

**Keywords:** Hanoi University of Pharmacy, postgraduate, pharmacy, satisfaction, Vietnam

## Abstract

The study aimed to investigate the determinants of pharmacy postgraduates’ satisfaction to suggest essential recommendations to enhance educational quality at Hanoi University of Pharmacy in Vietnam. A cross-sectional survey was conducted among 202 postgraduates using structured questionnaires, with 31 multidimensional questions and 1 question addressing overall satisfaction. Cronbach’s alpha was used to evaluate the questionnaire’s internal consistency, Exploratory Factor Analysis (EFA) identified key factors, and multiple linear regression analysis was applied to assess the impact of these factors. Overall satisfaction had a high mean score of 4.49 out of 5 (SD = 0.602). The final 29 questions were retained and divided into four main factors affecting satisfaction after rotating EFA. The dimension of “Support and Evaluation” was the most influential factor (β = 0.475), followed by “Training Organization”, “Facilities and Library”, and “Lecturers and Curriculum”. The variables with the lowest scores needed much more attention, including digital systems (4.2, SD = 0.852), information technology support (4.32, SD = 0.772), activities boosting lifelong learning skills (4.48, SD = 0.624), and the reasonability of the education program’s structure (4.48, SD = 0.608) and the studying program’s schedule (4.45, SD = 0.607). The findings indicate the issues that should be addressed, and have highlighted that improvements in electronic library accessibility and updated curricula are also recommended to further optimize the educational experience of postgraduate students.

## 1. Introduction

Postgraduate education is integral to advancing knowledge, fostering innovation, and preparing a highly skilled workforce to meet the demands of an increasingly complex and knowledge-driven global economy [1]. Postgraduate education is significant in pharmacy, as it equips pharmacy graduates with advanced and updated knowledge and skills in pharmaceutical sciences, practice, and legal regulations [2]. Therefore, ensuring the quality of postgraduate education is necessary for institutional success and is a critical factor ensuring the professional competencies of future experts and leaders [3]. One vital key point ensuring the quality of postgraduate education is student satisfaction, which has also served as a critical performance indicator, providing valuable insights into the effectiveness of educational programs and their capacity to address learners’ diverse needs and expectations [4]. This indicator was a multidimensional construct comprising administrative support, curriculum design, faculty quality, learning resources [5], the clear communication of academic regulations, evaluation criteria, comprehensiveness of course information [6,7], and respectfulness and responsiveness of administrative staff, combined with the timely resolution of student concerns [8], trust of students [4], the logical arrangement of courses and modules [9,10], well-equipped libraries, updated materials, accessible e-library systems [11], advanced laboratory facilities and academic technology [12], clean and conducive learning environments [13], and other dimensions [14,15,16]. Because of the multidimensionality and higher requirements for experimental and professional practices compared to other fields, developments and innovations in pharmacy education are difficult to carry out [17]. However, the COVID-19 pandemic had a profound impact, resulting in many pharmacy universities implementing major changes relating to infrastructure and course delivery within a short timeframe [18]. This transformation may have caused dissatisfaction and reduced the quality of education. Therefore, the evaluation of the satisfaction of postgraduates in this new context was necessary.

Several theoretical frameworks, including SERVQUAL [19], SERVPERF [20], and the Customer Satisfaction Index (CSI) [21,22], were developed with robust foundations for analyzing satisfaction and capturing service quality, performance perceptions, and customer loyalty. From these explorative factors provided, many studies on student satisfaction were conducted, adapted to each facility to understand and enhance student satisfaction. However, several research gaps remain in the COVID-19 context of postgraduate pharmacy education, where challenges have transformed the educational landscape, particularly regarding technological integration and virtual learning environments [23], highlighting the need for focused studies in this specialized field. These shifts have significantly influenced student expectations, yet their impact on postgraduate satisfaction, especially in specialized programs like pharmacy, has not been adequately studied. Addressing these gaps is essential to develop a holistic understanding of the unique factors influencing postgraduate satisfaction in pharmacy education. A comprehensive understanding of postgraduates’ satisfaction, motivation, and career aspirations is fundamental for developing a well-prepared and highly qualified pharmacy workforce [24]. It also provides a pathway for improving the quality and relevance of training programs in developing countries such as Vietnam.

The pharmacy postgraduate education system in Vietnam consists of educational levels including doctoral and master’s degrees and First-Level Specialized Pharmacist and Second-Level Specialized Pharmacist qualification. Among the four levels, the First-Level Specialized Pharmacist level belongs to the practical field instead of the academic field, like the master’s degree, and it is considered nearly equivalent a master’s degree with a time of completion of two years; similarly, the Second-Level Specialized Pharmacist level is in the practical field and nearly considered equivalent to a doctoral degree. Hanoi University of Pharmacy (HUP) is one of the leading universities in pharmacy education in Vietnam, with nearly 750 undergraduates with high levels of understanding and awareness [25] and 400 postgraduates each year. The study aimed to investigate the determinants of postgraduate student satisfaction at HUP, focusing on dimensions including support, evaluation, training organization, infrastructure, learning resources, lecturers, and the curriculum. These dimensions were considered based on the pharmacy context in Vietnam. By examining these factors, this study explores actionable insights to enhance educational quality according to the evolving needs of its postgraduate students, suggesting essential recommendations for similar universities not only in Vietnam but also in other countries with a similar context.

## 2. Materials and Methods

### 2.1. Participants

This study involved postgraduate students from HUP, including various specialization groups targeted to Master of Science (MS) and the First Level Specialized Pharmacist (FLSP) paths. The study incorporated two groups split as follows: the first group consists of post-graduate students who had recently completed their academic programs and were visiting HUP for transcript signing verification before graduation, which coincided with the time the survey was conducted; and the second group consists of students who graduated one year prior and had completed the transcript signing process before. A convenience sampling method was used, incorporating offline and online surveys from July 2023 to December 2023 to ensure inclusivity and broad accessibility.

An offline survey was conducted among the first group. Participants were given 5–10 min to complete the questionnaire on paper, after which, responses were carefully reviewed for completeness. Any incomplete or inconsistent answers were then discussed with the participants for confirmation. Only fully completed questionnaires were included in the analysis. An online survey via Google Forms was distributed to the second group.

According to Hair et al. [26], a minimum of five respondents per observed variable was necessary for Exploratory Factor Analysis (EFA). Given that this study included 31 observed variables, the required minimum sample size was 155 participants [26,27,28]. A total of 202 valid responses were collected, comprising 157 offline responses obtained from 222 distributed questionnaires and 45 online responses gathered from 67 online surveys received. This dataset was deemed appropriate for subsequent analyses, including descriptive statistics, exploratory factor analysis, hypothesis testing, and multiple regression analysis, ensuring the comprehensive evaluation of postgraduate student satisfaction.

### 2.2. Questionnaire Development

The survey instrument was designed based on a literature review with a total of 31 key questions selected and refined to comprehensively assess postgraduate student satisfaction. The questionnaire was initially drafted in English and later translated into Vietnamese through a meticulous process of translation and back-translation. This process was conducted by two bilingual experts in pharmacy education and one linguistics specialist to ensure both linguistic accuracy and conceptual equivalence. Any discrepancies identified during back-translation were resolved through discussion, and final adjustments were made to improve clarity and contextual appropriateness. To assess the validity and reliability of the questionnaire, a pilot study was conducted with 20 postgraduate students from Hanoi University of Pharmacy. Those postgraduate students were conveniently selected for the pilot study when they had just finished defending their thesis, with 3 months remaining before graduation. The instrument’s internal consistency was measured using Cronbach’s alpha, with an acceptable threshold set at 0.7 [27]. Based on the feedback, adjustments were made to refine the questionnaire before its final deployment. The survey consisted of mandatory items formatted on a 5-point Likert scale to assess various dimensions of satisfaction and factors influencing the postgraduate training experience.

### 2.3. Research Ethics

The purpose and content of the study were clearly explained to all participants before entering. They were requested to sign a voluntary informed consent. All collected information was encoded and kept confidential, and the information of participants was concealed and could not be identified in any way. The study was approved by the Ethics Review Committee under No 2306/PCT-HDDD, dated 5 June 2023, and the survey was carried out from July to December 2023 at HUP. Participation did not have any benefit or incentive.

### 2.4. Data Analysis

The collected data were analyzed using R version 4.4.2. Descriptive statistics were calculated for the demographic variables and response trends. Reliability testing was conducted using Cronbach’s alpha, with values between 0.8 and 1 considered excellent, 0.7–0.8 acceptable, and above 0.6 suitable for exploratory studies. Items with item–total correlations below 0.3 were excluded to maintain scale reliability [29].

EFA was performed to identify underlying factors influencing postgraduate student satisfaction. The Kaiser–Meyer–Olkin (KMO) measure and Bartlett’s test of sphericity confirmed the dataset’s suitability for factor analysis, with KMO values between 0.5 and 1 indicating appropriateness [26,28]. Factors with an eigenvalue greater than 1 were retained, and a cumulative variance above 50% was considered adequate. Items with factor loadings greater than 0.5 were included, while orthogonal varimax rotation was applied to clarify the component structure [29].

Multiple regression analysis was employed to examine the relationships among factors and their influence on student satisfaction. The model’s fit was evaluated using R^2^, F-statistics, and beta coefficients. A *p*-value less than 0.05 was considered significant.

## 3. Results

### 3.1. Demographics of Participants

The study collected 202 valid responses from postgraduate pharmacy students, comprising 157 offline and 45 online questionnaires. Table 1 presents the demographic data of the participants in terms of gender, age group, and specialization. The majority of respondents were female (65.3%), and most participants were aged between 18 and 35 years old (64.9%). Regarding specialized field, Pharmacology—Clinical Pharmacy (48.0%) and Pharmaceutical Management and Economics (38.1%) were the two most frequent specializations, reflecting a primary focus on these fields in the postgraduate programs.

### 3.2. Cronbach Alpha and Descriptive Statistics

The reliability analysis indicated that all factors satisfied the necessary thresholds, reflecting robust internal consistency. Table 2 shows Cronbach’s alpha for each dimension. Cronbach’s alpha values for all factors exceeded 0.6, and the item–total correlation coefficients were greater than 0.3, thereby confirming the reliability of the scales. Facilities, Equipment, and Library exhibited a Cronbach alpha of 0.897 based on seven observed variables, whereas Training Programs achieved a score of 0.898 with five observed variables. Training Organization exhibited a reliability coefficient of 0.890 across four variables, while the Lecturers factor attained a coefficient of 0.900 with six variables. Assessment and Evaluation achieved a score of 0.864 with three variables, while Student Support Activities exhibited the highest reliability at 0.911 with six variables. The results demonstrate that all factors exhibit reliability.

The data were suitable for factor analysis after the first round of EFA provided a KMO value of 0.950 and a Bartlett’s test *p*-value less than 0.05. The rotated factor matrix found four main factors with 29 variables fulfilling the criteria. The variables of “Appropriate time for transmitting course content” and “Training program content meets professional development requirements” were eliminated due to their factor loadings being below 0.5. To determine the optimal number of factors to retain, the scree plot was considered. The scree plot in Figure 1 reveals a clear “elbow” formation at the fourth component, supporting the retention of four factors.

After deleting these factors, a second round of EFA retained 29 variables, as shown in Table 3 and the Variance Explanation of Factors in Table 4.

### 3.3. Mean Scores of Variables

Table 5 presents the mean scores of postgraduate students’ satisfaction on each item. Overall, the satisfaction levels were relatively high, with mean scores ranging from 4.21 to 4.69 out of 5, and a mean score of general satisfaction of 4.49 (SD = 0.602). The lowest scores belonged to the factor of “Facilities and Library” comprising “the IT systems supporting learning and research” (4.32) and “the accessibility of the electronic library system” (4.21). However, “faculty members have strong teaching expertise and competence” received the highest rating (4.69), followed by “Course content is fully communicated to students” and “The lecturers provide comprehensive information about the course’s objectives, requirements, evaluation criteria, and reference materials” with similar scores (4.60) and the same factor (“Training Organization”).

### 3.4. Regression Results

The study employed multiple regression analysis to assess the impact of four identified factors on postgraduate students’ satisfaction. The findings from the multiple linear regression analysis, which emphasize the impact of these factors on overall satisfaction, are presented in Table 6.

The regression analysis results demonstrate that all *p*-values (Sig.) are below 0.05, indicating that the four factors significantly affect postgraduate students’ satisfaction with the training activities at Hanoi University of Pharmacy. The R^2^ value of 0.574 indicates that these factors account for 57.4% of the variance in satisfaction. Furthermore, all beta coefficients are positive, signifying that these factors positively influence students’ satisfaction.Y = 4.572 + 0.475 × X1 + 0.382 × X2 + 0.328 × X3 + 0.315 × X4

## 4. Discussion

With a total of 31 initial questions, the final 29 questions were retained and divided into four main factors that affected postgraduate student satisfaction. All scores were above 4, indicating high satisfaction among postgraduates. Although the six predefined categories to identify underlying dimensions influencing postgraduate satisfaction, the final results concentrated on a four-factor matrix, which was consistent with other studies [30,31]. This finding reinforced that postgraduates’ satisfaction was shaped by multiple dimensions and tended to cluster around key areas, even across different contexts. By addressing these, universities could boost the learners’ satisfaction, not only for undergraduates but also postgraduates. Addressing gaps in digital resource accessibility, faculty guidance, and curriculum alignment with pharmacy field needs could further enhance the academic experience and overall satisfaction.

The results showed that the mean values of satisfaction of all items ranged narrowly from 4.21 to 4.69, indicating a generally high level of satisfaction among respondents. These results reflected positively on the university’s quality and performance across various aspects, but the narrow range posed a challenge in clearly distinguishing which factors were perceived more positively than others. Therefore, while the results are encouraging, further studies using more discriminative scales or qualitative feedback may provide deeper insights into areas needing improvement.

Due to the digital transformation following the COVID-19 pandemic, many investments in e-learning systems were developed. However, the variable “The electronic library system is easily accessible for searching materials” had the lowest score (4.21), and the variable “The IT systems (learning devices, training management software, etc.) effectively support learning and research activities” had the second-lowest score (4.32) among the 29 variables. In other words, accessibility to electronic library resources remained challenging for students, and the effectiveness of IT systems in supporting learning and research activities was not fully optimized, despite significant investments in e-learning infrastructure. The benefits of e-learning were reported with higher satisfaction than traditional learning [32,33], but the shortcomings remained, and further studies to evaluate the real situation and problems relating to digital systems are necessary. Furthermore, this highlights the need for universities to improve their electronic library systems’ usability, upgrade digital platforms, expand available resources, and provide better technical support to ensure that students can fully benefit from digital academic resources because of the higher demands of postgraduate learners for researching academic references and the increasing reliance on digital learning tools [34,35] among postgraduates who are simultaneously working and learning.

The multiple regression model indicated that the model explained 57.4% of the variation in the dependent variables, meaning that the factors in the model had a fairly good level of influence. Among the four factors, the factor “Support and Evaluation” emerged as the most influential coefficient, significantly influencing satisfaction with postgraduate training at HUP. In this dimension, the “Academic support staff are responsible in their work” variable had the highest satisfaction score (4.56), showing the high appreciation of postgraduate students of the academic support staff. Although postgraduate students have diverse backgrounds, work experience, and personal lives, the academic support staff at HUP still met each student’s needs. The staff not only provided knowledge but also helped with academic issues and research orientation advice. According to the results of some studies, satisfaction with major areas of study is among the most essential factors influencing research achievement, preventing academic burnout and dropping out of learning early [36,37,38]. Moreover, good academic support can help postgraduate students overcome the challenges of adapting to a new learning environment, especially for those who were away from the academic environment for a long time and returned to study in an environment of high pressure as postgraduates [39,40]. Therefore, universities should continuously invest in professional development programs to enhance staff abilities and broaden personalized academic advising and mentoring projects or academic short courses to address the diverse needs of postgraduate students, particularly those who face difficulties in transitioning back into an intensive academic setting.

Also, among this dimension, the variable “Teaching and learning activities encourage students to enhance lifelong learning skills” had the lowest score (4.48). Although in higher education the primary responsibility for acquiring knowledge and skills falls on the students, the professors play a role in supporting learning by offering guidance, delivering constructive feedback, and providing academic support in order to foster self-education [41], and the expectations of postgraduates may be higher. Many studies also showed that university teachers faced significant challenges in providing guidance and academic supervision to optimize their students’ learning [41,42,43]. This implied that there might be a gap between postgraduate students’ needs and the level of academic support universities provide. Addressing these challenges may require universities to enhance learning skill development programs, adopt more student-centered teaching strategies, and implement support mechanisms to better align with postgraduate learners’ needs.

Some problems relating to the programs with the lowest scores should be addressed, including “The program structure ensures coherence and continuity with a balanced proportion of foundational and specialized knowledge” in the “Lecturer and Curriculum” dimension and “The arrangement of courses in the program is logical in sequence and timing” in the “Training Organization” dimension. These results indicate that postgraduate students felt inconsistencies in course structuring and studying at HUP came with a fairly high pressure, as well as the illogical arrangement of programs, which could affect students’ ability to build a solid specialized knowledge base and advance seamlessly through their studies. This result was similar to the previous results that showed learners’ satisfaction was affected by the course design [34,35]. Therefore, to address these issues, universities should conduct regular curriculum revisions and reviews to ensure that academic programs are well-structured, coherent, continuity, and aligned with students’ academic and professional development needs. Additionally, gathering continuous feedback can help refine course scheduling, ensuring that it accommodates learning progression and optimizes the program design. By improving curriculum coherence and course organization, institutions can enhance the overall learning experience and better support postgraduate students in achieving their academic goals.

The study had some limitations. First, the initial questionnaire had six factors, and only four factors were identified, so there may be other factors influencing the satisfaction of postgraduates that were unexplored, or some of the original factors may overlap and may only be part of a larger factor. Therefore, a further systematic review was vital for defining more factors. Second, the study focused on postgraduates, and as the sample size collection was conducted during their last visits to HUP for transcript signing before graduation, some postgraduates might have felt uncomfortable and concerned about evaluating the university and faculty on that day, so the sample size may not have been as populous and diversified. Therefore, we applied the anonymous questionnaire, and postgraduates submitted their answer by dropping them into the ballot box after completing the questionnaires to limit the discomfort and concerns of participants. Further studies with an extended sample size and diverse years of graduation are necessary. Third, the built programs and the requirements for the MS and FLSP cohorts differed and some concerns emerged relating to the different responses to the certain items. However, our study aimed to evaluate satisfaction and identify the determinants of postgraduate student satisfaction at HUP in general. So, though the programs and the requirements were different, they had to meet the needs of the learners and ensure the quality of postgraduate education. The responses of the postgraduate students focused on the quality of the programs and the requirements, in general, and this was also the main outcome that our study was aiming for, so the effect or bias between the respondents were limited. Fourth, the years of working experience were not collected in the demographic background data, which could be an influential variable in the results. Last but not least, the study chose a convenience sampling approach, which might inherently limit the generalizability due to potential sampling bias, though this method may be practical and easy to conduct among cross-sectional studies.

## 5. Conclusions

The results have demonstrated that there are four main factors that affected postgraduate student satisfaction. They include Support and Evaluation, Training Organization, Facilities and Library, and Lecturers and Curriculum. The findings highlight the critical role of administrative support, curriculum organization, and infrastructure in enhancing student satisfaction. Improvements in electronic library accessibility and updated curricula are recommended to further optimize the educational experience for postgraduate students at HUP.

## Figures and Tables

**Figure 1 pharmacy-13-00062-f001:**
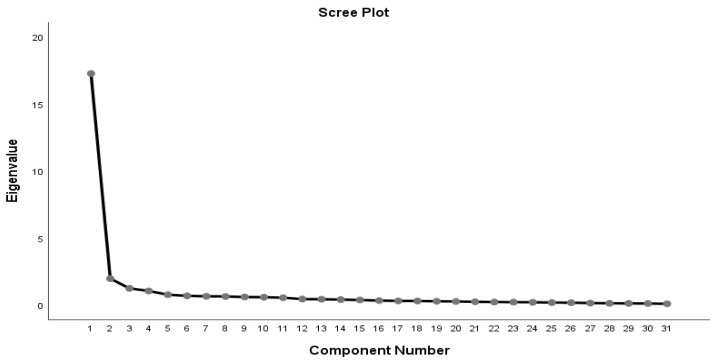
Scree plot.

**Table 1 pharmacy-13-00062-t001:** Demographic background of built environment students.

Items		Number (%)
Age (years)	18–35	131 (64.9%)
	36–50	71 (35.1%)
	≥51	0 (0%)
Gender	Male	70 (34.7%)
	Female	132 (65.3%)
Specialized courses	Pharmaceutics and Pharmaceutical Technology	3 (1.5%)
	Pharmacognosy and Traditional Medicine	8 (4.0%)
	Pharmacology—Clinical Pharmacy	97 (48.0%)
	Pharmaceutical Biochemistry	5 (2.5%)
	Drug Quality Control and Toxicology	12 (5.9%)
	Pharmaceutical Management and Economics	77 (38.1%)

**Table 2 pharmacy-13-00062-t002:** Cronbach’s alpha coefficient values for each factor.

No	Factor	Number of Variables	Alpha Cronbach Coefficient
1	Facilities, Equipment, and Library	7	0.897
2	Training Programs	5	0.898
3	Training Organization	4	0.890
4	Lecturers	6	0.900
5	Assessment and Evaluation of Learning Outcomes	3	0.864
6	Student Support Activities	6	0.911

**Table 3 pharmacy-13-00062-t003:** Final factor loading.

Items	Factors			
	Support and Evaluation	Training Organization	Facilities and Library	Lecturers and Curriculum
Evaluating regulations are clearly explained to students	0.718			
Administrative staff are polite, gentle, and respectful attitude toward students	0.693			
Administrative procedures are resolved promptly and appropriately	0.674			
Assessment test results accurately reflect students’ abilities	0.674			
Learning environment is guaranteed and protected	0.669			
Learners’ appropriate demand is promptly resolved	0.651			
Learners’ research activities are effectively supported	0.647			
Administrative staff are responsible	0.634			
Teaching activities encourage lifelong learning	0.626			
Methods of assessing students’ results ensure validity, reliability, and fairness	0.577			
Academic regulations are clearly explained to students		0.726		
Module content is fully explained to students		0.700		
Students are provided with complete modules (purpose, requirements, evaluations, and reference documents)		0.693		
Exam schedule is arranged with student appropriately notified		0.662		
Knowledge provided in the program is advanced compared to that at the university’s program		0.508		
Courses in the program are arranged in a logical sequence and at the appropriate time		0.506		
Technology system (learning technology, academic software…) supports learning and research activities			0.751	
Library is fully equipped to meet the academic demands			0.739	
Facilities and appliances in laboratories meet the academic demands			0.738	
It is convenient to access and search resources in the e-lib			0.730	
Accessible and searchable online library			0.701	
Learning environment is clean			0.687	
Library materials are updated regularly			0.543	
Lecturers are highly skilled in teaching				0.741
Students are guided in how to study effectively				0.679
Structure of the academic program is balanced between teaching parts				0.580
Teaching content is useful to students				0.559
Students are inspired by lecturers				0.552
Students are proactive with positive teaching activities				0.543
KMO	0.950			
*p*-value in Bartlett test	<0.001			

**Table 4 pharmacy-13-00062-t004:** Variance explanation of factors.

Factor	Eigenvalues			Total Variance Explained			Cumulative Variance Explained After Rotation		
	Total	% Variance	Cumulative % Variance	Total	% Variance	Cumulative % Variance	Total	% Variance	Cumulative % Variance
1	17,276	55,729	55,729	17,276	55,729	55,729	6562	21,169	21,169
2	1973	6364	62,093	1973	6364	62,093	5876	18,955	40,123
3	1245	4015	66,108	1245	4015	66,108	5012	16,169	56,293
4	1048	3379	69,487	1048	3379	69,487	4090	13,194	69,487

**Table 5 pharmacy-13-00062-t005:** Mean scores of items.

Factors	Item	Mean	Standard Deviation
Support and evaluation	Evaluating regulations are clearly explained to students	4.53	0.548
	Administrative staff are polite, gentle, and respectful attitude toward students	4.52	0.583
	Administrative procedures are resolved promptly and appropriately	4.45	0.661
	Assessment test results accurately reflect students’ abilities	4.51	0.584
	Learning environment is guaranteed and protected	4.50	0.566
	Learner’s appropriate demand is promptly resolved	4.53	0.566
	Learner’s research activities are effectively supported	4.52	0.600
	Administrative staff are responsible	4.56	0.554
	Teaching activities encourage lifelong learning	4.48	0.624
	Methods of assessing students’ results ensured validity, reliability and fairness	4.54	0.556
Training organization	Academic regulations are clearly explained to students	4.53	0.548
	Module content is fully explained to students	4.60	0.521
	Students are provided with complete modules (purposes, requirements, evaluations, reference documents)	4.60	0.521
	Exam schedule is arranged with student appropriately notified	4.55	0.581
	Knowledge provided in the program is advanced compared to that at the university’s program	4.45	0.607
	Courses in the program are arranged in a logical sequence and at the appropriate time	4.52	0.566
Facilities and library	Technology system (learning technology, academic software…) supports learning and researching activities	4.32	0.772
	Library is fully equipped to meet the academic demands	4.36	0.692
	Facilities and appliances in laboratories meet the academic demands	4.40	0.663
	It is convenient to access and search the resource in the e-lib	4.37	0.673
	Accessible and searchable online library	4.21	0.852
	Learning environment is clean	4.49	0.600
	Library’s materials are updated regularly	4.31	0.703
Lecturers and curriculum	Lecturers are highly skilled in teaching	4.69	0.494
	Students are guided in how to study effectively	4.56	0.545
	Structure of the academic program is balanced between teaching parts	4.48	0.608
	Teaching content is useful to students	4.52	0.557
	Students are inspired by lecturers	4.59	0.532
	Students are proactive with positive teaching activities.	4.54	0.574
General satisfaction		4.49	0.602

**Table 6 pharmacy-13-00062-t006:** Coefficients.

Factor	Unstandardized Coefficients	Standard Coefficients	Sig.	VIF
Constant	4.572		0.000	
Support and Evaluation (X1)	0.260	0.475	0.000	1.000
Training Organization (X2)	0.208	0.382	0.000	1.000
Facilities and Library (X3)	0.178	0.328	0.000	1.000
Lecturers and Curriculum (X4)	0.171	0.315	0.000	1.000
Sig. F	0.000			
Adjusted R Squared	0.574			

## Data Availability

Data supporting the reported results can be provided by the corresponding author upon reasonable request.

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
