# Peer review of "Influential Factors on Postgraduates’ Satisfaction with the Quality of Pharmacy Education: Evidence from a University in Vietnam"

_pharmacy, 2025, doi:10.3390/pharmacy13030062_

Round 1

Reviewer 1 Report

Comments and Suggestions for Authors

I applaud the authors for their hard work and dedication in conducting this research and understanding the importance of student feedback in improving the quality of education. 

Please find my general comments below 

  1. It is not directly clear how many years post-grad the participants were.
  2. Was there any incentive for completing the survey
  3. Were the individuals who completed this prior you signing their transcripts mandated to complete the survey?
  4. The authors indicate that they utilized both thr MSi and FLSP cohorts. Do these cohorts have a different program/requirements that would affect how they responded to certain items?
  5. Your limitations section describes how you overcame limitations, which is great for readers. However it does not list the specific limitations of your study that could impact results. 

Author Response

Reviewer 1

Thank you so much for all the meaningful comments of the reviewer

Comments and Suggestions for Authors

I applaud the authors for their hard work and dedication in conducting this research and understanding the importance of student feedback in improving the quality of education. 

Please find my general comments below 

  1. It is not directly clear how many years post-grad the participants were.

Response: Thank you for your comment. We have added the studying years of postgraduate students in the introduction section, Lines 80-81.

  1. Was there any incentive for completing the survey

Response: Thank you for your valuable comment. Participation did not have any benefit or incentive. We have added this information for clearer (Materials and Methods part, Line 141).

  1. Were the individuals who completed this prior you signing their transcripts mandated to complete the survey?

Response: Thank you. The survey was optional for them. The transcripts and degrees were signed and completed already prior when individuals responded the survey. We have revised the sentence for clearer (Lines 96-100)

  1. The authors indicate that they utilized both thr MSi and FLSP cohorts. Do these cohorts have a different program/requirement that would affect how they responded to certain items?

Response: Thank you for your insightful comment. As you commented, the built programs and the requirements for the MS and FLSP cohorts differed. However, our study aimed to evaluate satisfaction and identify the determinants of postgraduate student satisfaction at HUP in general. So, though the programs and the requirements were different, they had to take studying persons-centers, meet the needs of the learners, and ensure the quality of postgraduate education. The responses of the postgraduate students were the quality of the programs and the requirements in general, and this was also the main outcome that our study wanted to aim for, so effect or bias between the respondents were limited. We also added this discussion as the limitation into the manuscript (Discussion part, line 323-328)

  1. Your limitations section describes how you overcame limitations, which is great for readers. However it does not list the specific limitations of your study that could impact results.

Response: Thank you very much for your helpful comment. We truly appreciate your suggestion to clarify and highlight the limitations that may have impacted our study results. We have revised the limitations with the relevant solutions stated clearly (Lines 308-332).

Reviewer 2 Report

Comments and Suggestions for Authors

Dear authors:

This paper describes a study on the satisfaction of postgraduate students at a university in Vietnam. I believe it may be of interest to professionals in the pharmaceutical field, especially those involved in the training of pharmacists. Before publication, you should consider certain aspects, which I list briefly below:

  • In the title, they could add at the end: a case study at a university.

  • On line 13, the semicolon (;) should be changed to a period (.) and I would include, after "quality", the phrase at a university in Vietnam (or even include the name of the university).

  • On line 18, change the semicolon to a period.

  • Also on line 18, when indicating the mean value, it should be noted that it is very high.

  • On line 23, clarify what "IT" stands for.

  • On page 101, it should be stated whether the postgraduates completed the form on paper or online.

  • On line 102, it is not specified whether the survey was anonymous or not.

  • On line 103, could the number of rejected questionnaires versus the total be indicated?

  • In the same paragraph, it seems that only the opinions of recent graduates are collected, not of those who graduated years earlier: does this not affect the representativeness of the results?

  • On line 104, the phrase in the second group is repeated twice.

  • Part of this information is repeated on lines 294 to 296; it could be mentioned only at the beginning, indicating, as is later said, that the response is neither popular nor diverse.

  • Lines 122 to 124: How were those 20 postgraduates selected?

  • In Table 5, all the mean values range between 4.2 and 4.6, which makes it difficult to distinguish between the different items. This should be discussed in the explanatory text.

  • Lines 212 to 213: the text does not explain whether there is a difference based on years of experience.

  • The limitations discussed on lines 290 to 295 should be included in the introduction and explained in more detail, as they limit the conclusions drawn from the study.

  • The conclusions are too brief and should be elaborated further.

  • In some references, such as 12, 20, 38, 40, 41, and 44, there are acronyms whose meanings are unclear.

In summary, I believe this is a text that would be of interest to pharmacy professionals concerned with university training programs, but it needs improvement in the areas mentioned above, as well as any others that might be identified by additional reviewers.

All the best

Author Response

Reviewer 2

Thank you so much for all the meaningful comments from the reviewer

Dear authors:

This paper describes a study on the satisfaction of postgraduate students at a university in Vietnam. I believe it may be of interest to professionals in the pharmaceutical field, especially those involved in the training of pharmacists. Before publication, you should consider certain aspects, which I list briefly below:

  • In the title, they could add at the end: a case study at a university.

Response: Thank you. However, we suggested replacing “a case study” with “evidence” to avoid misunderstanding the type of study design of “the case study”.

  • On line 13, the semicolon (;) should be changed to a period (.) and I would include, after "quality", the phrase at a university in Vietnam (or even include the name of the university).

Response: Thank you. We have revised.

  • On line 18, change the semicolon to a period.

Response: Thank you. We have changed.

  • Also on line 18, when indicating the mean value, it should be noted that it is very high.

Response: Thank you. We have revised.

  • On line 23, clarify what "IT" stands for.

Response: Thank you. We have clarified.

  • On page 101, it should be stated whether the postgraduates completed the form on paper or online.

Response: Thank you. We have clarified and revised the Materials and Methods section, Lines 104.

  • On line 102, it is not specified whether the survey was anonymous or not.

Response: Thank you for your valuable comment. All the information of participants was anonymous. We have added this information to the Research Ethics section, Lines 138-140.

  • On line 103, could the number of rejected questionnaires versus the total be indicated?

Response: Thank you for your comment. This was indicated in the Lines of 111-114.

  • In the same paragraph, it seems that only the opinions of recent graduates are collected, not of those who graduated years earlier: does this not affect the representativeness of the results?

Response: Thank you for your thoughtful comment. We have clarified and revised the paragraph for choosing participants (Lines 96-100). It ensured partly the representativeness of the results.

  • On line 104, the phrase in the second group is repeated twice.

Response: Thank you. We have deleted.

  • Part of this information is repeated on lines 294 to 296; it could be mentioned only at the beginning, indicating, as is later said, that the response is neither popular nor diverse.

Response: Thank you for your insightful comment. We have revised this paragraph (Lines 313-320).

  • Lines 122 to 124: How were those 20 postgraduates selected?

Response: Thank you for your valuable comment. The 20 postgraduate students were selected conveniently when they had just finished the thesis defending and had 3 months before the graduation event for pitot study. We have added this into the manuscript, Materials and Methods section, Lines 128-130.

  • In Table 5, all the mean values range between 4.2 and 4.6, which makes it difficult to distinguish between the different items. This should be discussed in the explanatory text.

Response: Thank you for your comment. We have discussed more in Discusion section, line 235-241.

  • Lines 212 to 213: the text does not explain whether there is a difference based on years of experience.
  • Response: Thank you for your comment. The years of working experience did not collected in demographic background. This is one of our limitation, and we have added in the Discussion section, line 329-330
  • The limitations discussed on lines 290 to 295 should be included in the introduction and explained in more detail, as they limit the conclusions drawn from the study.

Response: Thank you so much for your valuable comment. We completely agree that the limitations discussed on lines 290-295 could influence the interpretation of the study’s conclusions drawn from the study and including them in the introduction could support the deep analysis and insightful explanation, as well as provide a more comprehensive context for the readers. However, these limitations were withdrawn carefully and rigorously after completing the study, rather than being anticipated at the outset. Therefore, the introduction comprising them may cause unreasonably in the process of doing our study, and create an impression of methodological bias. For this reason, we believe it is more appropriate to keep this part in the Discussion section, where their impact on the findings is analyzed in context. instead of in the Introduction section.

  • The conclusions are too brief and should be elaborated further.

Response: Thank you. We have elaborated further.

  • In some references, such as 12, 20, 38, 40, 41, and 44, there are acronyms whose meanings are unclear.

Response: Thank you so much. We have revised the references.

In summary, I believe this is a text that would be of interest to pharmacy professionals concerned with university training programs, but it needs improvement in the areas mentioned above, as well as any others that might be identified by additional reviewers.

All the best

Reviewer 3 Report

Comments and Suggestions for Authors

Dear authors, thank you for giving me the possibility to review the article “Influential factors on postgraduates' satisfaction with the qual-2 ity of pharmacy education in Vietnam”. The article is interesting from a scientific point of view, however for publication it would be useful to provide some more information: Specifically, in the methodology section:
It talks about sampling was by convenience, which limits generalization.
You should consider adding this limitation explicitly in the discussion.
In the results section:
1. indicate whether items with an Alpha less than .6 were eliminated in the process and how that affected the total alpha. You could also have used Alpha if Item Deleted to show the robustness of each item.
2.    In the factor analysis you state that Varimax rotation was used but there is no justification as to why or if another technique was used (oblique or orthogonal?). In addition, Eigenvalues are mentioned but it is not reported whether Kaiser criterion (≥1) or scree plot was used.
3.    In the descriptive analysis, I suggest adding confidence intervals and ANOVA between factors.

Author Response

Reviewer 3

Thank you so much for all the meaningful comments from the reviewer

Dear authors, thank you for giving me the possibility to review the article “Influential factors on postgraduates' satisfaction with the qual-2 ity of pharmacy education in Vietnam”. The article is interesting from a scientific point of view, however for publication it would be useful to provide some more information:

Specifically, in the methodology section: It talks about sampling was by convenience, which limits generalization. You should consider adding this limitation explicitly in the discussion.

Response: Thank you so much for your comment. We have added the limitation of convenience sampling method in the Discussion part, Lines 332-335.

In the results section:

  1. indicate whether items with an Alpha less than .6 were eliminated in the process and how that affected the total alpha. You could also have used Alpha if Item Deleted to show the robustness of each item.

Response: Thank you so much for your comment At the beginning of the analysis, all 31 items had Cronbach’s Alpha values greater than 0.6 , so no item was eliminated based on this criterion. However, during the first round of factor analysis, two items were removed due to having factor loadings lower than 0.5. Regarding the Alpha if Item Deleted values, we acknowledge this is a useful metric for assessing the robustness of each item, and these Alpha if Item Deleted were available on our output file. However, our study used the main factors to present the results, so we did not include the Alpha if Item Deleted for each item in the presentation of results. Nevertheless, we will send this information to readers concerned if requested to enhance the transparency and robustness of our findings.

  1. In the factor analysis you state that Varimax rotation was used but there is no justification as to why or if another technique was used (oblique or orthogonal?). In addition, Eigenvalues are mentioned but it is not reported whether Kaiser criterion (≥1) or scree plot was used.

Response: Thank you very much for your insightful comment. We used Varimax rotation in EFA when we assumed that in factors extracted were not correlated. In this situation, Varimax rotation was used as it appropriate. For the Varimax rotation and other perpendicular rotations, it was more suitable for Principal Component Analysis extraction when the main purpose is to narrow down the number of observed variables on the representative factors with the most extracted variance. However, we acknowledge your comment and we will consider testing both orthogonal and oblique rotations to ensure robustness and better theoretical alignment in future research. Regarding scree plot, we have added in the manuscript (Lines 188-191, and Figure 1).

  1. In the descriptive analysis, I suggest adding confidence intervals and ANOVA between factors.

Response: Thank you so much for your thoughtful suggestion. We agree that adding confidence intervals and conducting ANOVA could provide deeper results. However, this section aimed to inform practical improvements rather than make statistical comparisons between factors. To find out the factors influencing postgraduate students’ satisfaction was the main purpose of our study, and from then, to propose strategies and suggestions for quality improvement.  As such, we believe that including confidence intervals or ANOVA may not substantially enhance the interpretation of the results. We appreciate your comment and will keep this consideration in mind for further studies and future improvements.

Round 2

Reviewer 3 Report

Comments and Suggestions for Authors

The authors have adequately addressed all comments made during the review process. They have responded to each comment with clarity and have incorporated relevant suggestions, resulting in a significant improvement of the manuscript. I believe that the work has been strengthened and is ready to move forward in the editorial process.